# The Association of Carcinoembryonic Antigen (CEA) and Air Pollutants—A Population-Based Study

Yi-Jen Fang [1,2], Lukas Jyuhn-Hsiarn Lee [1,3,4], Kuei-Hau Luo [5], Po-Sheng Fang [6], Chen-Cheng Yang [7] and Hung-Yi Chuang [1,5,8,9,*]

1. Ph.D. Program in Environmental and Occupational Medicine, College of Medicine, Kaohsiung Medical University, Kaohsiung City 807, Taiwan; fang531109@gmail.com (Y.-J.F.); lukaslee@nhri.edu.tw (L.J.-H.L.)
2. Digestive Disease Center, Show-Chwan Memorial Hospital, Changhua 500, Taiwan
3. National Institute of Environmental Health Sciences, National Health Research Institutes, Miaoli City 350, Taiwan
4. Institute of Environmental and Occupational Health Sciences, College of Public Health, National Taiwan University, Taipei 100, Taiwan
5. Graduate Institute of Medicine, College of Medicine, Kaohsiung Medical University, Kaohsiung City 807, Taiwan; u107800007@kmu.edu.tw
6. School of Medicine, College of Medicine, Kaohsiung Medical University, Kaohsiung City 807, Taiwan; u105001067@kmu.edu.tw
7. Department of Occupational and Environmental Medicine, Kaohsiung Municipal Siaogang Hospital, Kaohsiung City 812, Taiwan; u106800001@kmu.edu.tw
8. Department of Occupational and Environmental Medicine, Kaohsiung Medical University Hospital, Kaohsiung City 807, Taiwan
9. Department of Public Health and Environmental Medicine, College of Medicine, Research Center for Environmental Medicine, Kaohsiung Medical University, Kaohsiung City 807, Taiwan
* Correspondence: ericch@kmu.edu.tw; Tel.: +886-7312-1101

**Abstract:** Air pollutants are substances in the air that have harmful effects on humans and the ecological environment. Although slight elevations in carcinoembryonic antigen (CEA) are commonly observed in apparently healthy persons, potential associations between CEA levels and chronic low-grade inflammation induced by air pollution have yet to be documented. We conducted a community-based cross-sectional study to estimate the association between short-term exposure to ambient air pollution and the CEA. A total of 9728 participants from health examinations were enrolled for the analysis and linked with their residential air pollutant data including ozone ($O_3$), nitrogen dioxide ($NO_2$), carbon monoxide (CO), sulphur dioxide ($SO_2$), and particulate matter ($PM_{10}$). The results showed that every increase of 1 ppm $O_3$ significantly increased the mean differences of the CEA blood concentration by 0.005 ng/mL. Each increase of 1 ppm CO significantly reduced the mean differences of the CEA blood concentration by 0.455 ng/mL. Although smoking and alcohol drinking also increased the CEA levels, with adjustment of these confounders we identified a significant association between serum CEA in the general population and levels of the air pollutants $O_3$ and CO. In conclusion, the serum CEA concentrations and short-term air pollutants $O_3$ and CO exposure were found to have a significant relationship; however, its mechanism is still unclear. Moreover, long-term air pollution exposure and changes in CEA concentration still need to be further evaluated.

**Keywords:** air pollutant; carcinoembryonic antigen (CEA); chronic inflammation

## 1. Introduction

Air pollutants are substances in the air that have harmful effects on humans and the ecological environment. The substances have various forms, which can be particles, droplets, or gases; they can be formed naturally or produced by human [1]. Air pollutants are divided into primary pollutants and secondary pollutants. The main pollutants are

usually produced by processes, such as fly ash from volcanic eruptions and industries. Others include carbon monoxide (CO) gas from motor vehicle exhaust, or sulphur dioxide ($SO_2$) released from industries that use or burn coal and petroleum containing sulphur. The oxides of nitrogen (mainly $NO_2$) are formed by various combustions, such as vehicles, industries, forest fires. Secondary pollutants are not emitted directly but are formed in the air when the main pollutants undergo reactions or interactions. Low-level ground ozone ($O_3$) is an example of secondary pollutants. Due to the influence of industrial and automobile exhaust gas, the photochemical smog produced by nitrogen dioxide in the exhaust of steam and locomotives, ground $O_3$ formed by NOx and volatile organic compounds through photochemical reaction, forms and accumulates on the surface, and the $O_3$ in the lower air is called for a harmful zone [1,2].

The gases $O_3$, CO, $SO_2$, $NO_2$, and particulate matter ($PM_{10}$ and $PM_{2.5}$) are the main ambient pollutants that are proven to have multiple adverse influences on health, including respiratory and cardiovascular diseases. Previous literature shows that exposure to air pollutants was associated with the risk of hospitalization, emergency visits, and mortality, even cancers [2–8].

Ambient carbon monoxide (CO) is one of the air pollutants. It is colourless, odourless, and tasteless that is primarily produced by incomplete combustion of fuels such as motor vehicle exhaust and industrial emission. Toxic effects of CO are mainly produced by its capacity to bind to and altering the function and metabolism of heme proteins. As the known formation of carboxyhaemoglobin decreases the $O_2$ carrying capacity and impairs releasing of $O_2$ from haemoglobin to its use in tissues. On the other hand, recent studies found that CO at lower concentrations can have anti-inflammatory effects [9–11]. We found some toxicologic and epidemiologic researchers had focused on examining the effects of low concentrations of CO (50 ppm and less) without overt CO poisoning, which could be therapeutically effective [12,13]. Epidemiological studies and animal studies have examined the associations between short-term exposure to CO and morbidity and mortality risk, but the results have been mixed and controversial [14–20].

The $PM_{10}$ is respirable particulate matter with an aerodynamic equivalent diameter of less than or equal to 10 microns. It is a general term for small solid or liquid particles suspended in a gas. Particulate matter is naturally produced and originates from volcanoes, sandstorms, forest, and grassland fires. Other human activities, such as burning fossil fuels in vehicles, power plants and various industrial processes, also produce a large volume of particulate pollutants. Others are caused by the environment. Fine particles formed by the interaction of sulphur oxides, nitrogen oxides, volatile organic compounds, and other compounds in the air. The chemical and physical composition of PM varies greatly depending on the location, climate, and season. It can be directly inhaled into the respiratory tract, but part of PM can be excreted through coughing sputum [1,21]. The PM with larger particles will also be blocked by the villi inside the nasal cavity. The increase in PM is associated with health hazards, such as changes in lung function, and lung cancer and heart disease [4,5,21].

Carcinoembryonic antigen (CEA) is a set of highly related glycoproteins, which is an embryonic tumour antigen produced in gastrointestinal tissue from the human embryonic stage to the foetus. In adults, CEA is mainly secreted by colonic mucosal cells, with a small amount of reabsorbed into the blood. Its half-life in blood is about two days [22]. Clinically CEA is used as one of tumour markers because it is expressed at greatly increased levels in adenocarcinomas, particularly in colorectal cancer, and cancers of pancreas, lung, prostate, ovary, and breast [23–26]. In addition, modestly increasing serum CEA levels may be associated to several non-neoplastic conditions such as renal and hepatic insufficiency, as well as aging; moreover, CEA levels can also be elevated in heavy smokers [24,25,27]. On the other hand, CEA elevation was also identified in hypothyroidism and chronic obstructive pulmonary disease, though at levels less than 10 ng/mL [27–29]. Furthermore, some studies have suggested that increased serum CEA were associated with metabolic syndrome, insulin resistance, carotid atherosclerosis, and type 2 diabetes, all of which are

conditions associated with oxidative stress and chronic low-grade inflammation [27–33]. However, research is rare that investigates the air pollutants influencing CEA in humans.

Serum CEA is sometimes checked at health examinations in Taiwan; however, potential associations between CEA and chronic low-grade inflammation induced by air pollution have yet to be documented. In addition, few studies to date have examined the relationship between serum CEA and air pollution in a large, population-based study. Thus, we conducted a community-based cross-sectional study to estimate the association between short-term exposure to ambient air pollution and the serum CEA levels.

## 2. Materials and Methods

This community-based cross-sectional study was established from January 2002 and December 2004 was sponsored by and based on the health promotion program of the Health Bureau of Kaohsiung Municipality. Kaohsiung Medical University and Kaohsiung Municipal Siaogang Hospital administered the health examinations. All the clinical laboratory examinations were performed in the central laboratory of Municipal Siaogang Hospital, which was accredited by Taiwan Accreditation Foundation, a member of International Laboratory Accreditation Cooperation. The study was conducted in accordance with the ethical principles of the Declaration of Helsinki and was reviewed and approved by the Institutional Review Board of Kaohsiung Medical University (KMUH-IRB990206).

The project initially planned to select 10,000 citizens from the metropolitan area of Kaohsiung City. Adult residents of Kaohsiung City, who were randomly sampled from 11 metropolitan districts and used stratified proportion of population of the randomly selected districts, were invited to participate in this health survey program by letters and telephone calls. Participants were volunteers who responded to the invitations, and a consent form was filled out for each participant.

Participants received a questionnaire survey that documented gender, age, smoking, drinking, and chewing betel nut. Body height and weight were measured by an electric anthropometer (HGM-300, CROWN®, Taipei City, Taiwan). Routine blood tests for white blood cells, red blood cells, hemoglobulin, and platelets, and general biochemical tests for liver function index, creatinine, CEA, and chest X-rays. Serum CEA levels were quantified by chemiluminescence immunoassay using a Unicel DXI 800 analyser (Beckman-Coulter, Chaska, MN, USA). We excluded participants who met at least one of the following criteria: missing data on questionnaire; a history of cancer; respiratory, renal, hepatobiliary, thyroid, or rheumatologic diseases. In addition, subjects with leukocyte counts <3000 cells/µL or >10,000 cells/µL, or those with CEA levels >10 µg/L were excluded to rule out possible cases of bone marrow abnormalities, infection, or hidden malignancy. Although these participants did not enter to data analysis, they all received a note to call back for follow-up visits in the hospitals. A total of 40 patients with known cancer-related diseases in the past medical history were excluded; the final participate number was 9728.

The air pollution was detected by the air monitoring stations. The air pollutants concentration was interpolated for the participant's address on the map from the nearest air monitoring station. Data for $PM_{10}$, CO, $O_3$, $NO_2$, and $SO_2$ were hourly collected at these monitoring stations, set up by the Taiwan Environmental Protection Administration. There was at least one air monitoring station in each district in the metropolitan area of Kaohsiung City. The detailed information can be referred to: https://airtw.epa.gov.tw/ENG/default.aspx (assessed on 6 March 2022). Considering that the average half-life of CEA is two days; therefore, the air pollution concentration measured on the day of the examination day and the day before the examination day were taken as the average of the air pollution concentrations and used as the exposure variables.

The continuous variables of the examinee including age, height, weight, BMI, were presented as means and standard deviation (SD). The value of CEA of 5 ng/mL was used as the normal value to divide the CEA into two groups, then 2 sample *t*-tests were used to examine whether $PM_{10}$, CO, $O_3$, $NO_2$, $SO_2$, age, BMI, WBC, AST, ALT, and Creatinine were different between the two CEA groups. The 2 sample *t*-test was used to evaluate

whether there was a difference in CEA level between smoking, drinking, chewing betel nut, and chest X-ray for lung problems. Pearson correlation analysis was used to evaluate the correlation between CEA and air pollution.

We used multiple linear regression to analyse the variables and established two sets of models for comparison. Model 1 evaluates the correlation between CO, $O_3$, $NO_2$, $SO_2$, age, gender, WBC, ALT, AST, chest X-ray, daily habit of tobacco smoking, alcohol, betel nut use status, and CEA. Model 2 was in addition to Model 1 and added $PM_{10}$ to evaluate the correlation with CEA to further explore the relationship between air pollution and CEA. Considering that air pollutant concentrations would be highly correlated, we estimated the variance inflation factor (VIF) as part of the model-building to explore the collinearity among these air pollutants. The α error was set as 0.05 two-tailed, and SPSS 20 was performed the statistical analysis (SPSS version 20.0, IBM Corp., Armonk, NY, USA)

## 3. Results

Table 1 shows the basic characteristics of health examinees. The number of examinees was 4188 (43.1%) males and 5540 (56.9%) females. Their average (±standard deviation) age was 54.3 (±5.9) years old. Their average height was 160.4 (±7.9) cm, their weight was 63.7 (±10.8) kg, and mean of the BMI was 24.7 (±3.8) kg/m$^2$. In routine blood tests, the average value of white blood cells was 5911.1 (±1477.8)/μL. The general biochemical value of AST was 29.5 (± 20.2) U/L; ALT was 28.2 (± 29.1) U/L. The average concentration of CEA in blood was 1.7 (±3.0) ng/mL. Chest X-ray examination showed that there were 2821 (29%) people with minor lung abnormalities, including minor fibrosis, small nodules (<3), mild lung marking increases, and 6902 (70.9%) people with normal or degeneration of the thoracic spine. There were 1456 (15%) people who smoked, 1382 (14.2%) people who had a habit to drink alcohol, and 273 (2.8%) people who ate betel nuts every day.

**Table 1.** Demographic characteristics of participants in the study.

| Characters | Mean ± SD or N (%) | IRQ (Q25–Q75) |
|---|---|---|
| Gender | | |
| Male | 4188 (43.1%) | |
| Female | 5540 (56.9%) | |
| Age (years) | 54.3 ± 5.9 | 49.4–59.2 |
| Height (cm) | 160.4 ± 7.9 | 154.8–165.9 |
| Weight (kg) | 63.7 ± 10.8 | 56.0–70.4 |
| Body Mass Index (kg/m$^2$) | 24.74 ± 3.79 | 22.4–26.7 |
| WBC (per μL) | 5911.1 ± 1477.8 | 4870–6730 |
| SBP (mmHg) | 128.2 ± 20.2 | 114–140 |
| DBP (mmHg) | 78.5 ± 12.0 | 70–86 |
| AST (U/L) | 29.5 ± 20.2 | 21–31 |
| ALT (U/L) | 28.2 ± 29.1 | 14–31 |
| Chest X ray (lung) | | |
|    Minor abnormal finding | 2821 (29.0%) | |
|    Normal | 6902 (70.9%) | |
| Tobacco | | |
|    Yes | 1456 (15%) | |
|    No | 8092 (83.2%) | |
| Alcohol | | |
|    Yes | 1382 (14.2%) | |
|    No | 8225 (84.5%) | |
| Betel nut | | |
|    Yes | 273 (2.8%) | |
|    No | 9349 (96.1%) | |
| CEA (ng/mL) | 1.62 ± 1.22 | 0.76–2.07 |
| Air pollution [a] | | |
|    PM10 (μg/m$^3$) | 68.6 ± 32.2 | 41.3–93.0 |
|    CO (ppm) | 0.6 ± 0.2 | 0.45–0.75 |
|    $O_3$ (ppm) | 28.2 ± 12.5 | 16.9–38.2 |
|    $NO_2$ (ppb) | 25.6 ± 12.4 | 16.4–31.5 |
|    $SO_2$ (ppb) | 8.2 ± 4.8 | 4.8–10.2 |

[a]: 2 days means (the day and previous day) from the nearest monitoring station.

Air pollution indicators. The average value of the two days before the health check were: $PM_{10}$ 68.6 ($\pm$32.2) ($\mu g/m^3$), CO 0.6 ($\pm$0.2) ppm, $O_3$ 28.2 ($\pm$12.5) (ppm), $NO_2$ 25.6 ($\pm$12.4) ppb, and $SO_2$ 8.2 ($\pm$4.8) ppb.

Table 2 is divided into two groups based on the CEA blood concentration: CEA $\leq$ 5 ng/mL ($n$ = 9476) and CEA > 5 ng/mL ($n$ = 252). In the CEA > 5 ng/mL group, the age, white blood cells, and AST ALT values were significantly higher than those in the CEA $\leq$ 5 ng/mL group ($p$-value < 0.05). In terms of air pollutants, the $NO_2$ concentration (25.7 $\pm$ 12.4) ppb and $PM_{10}$ concentration (68.7 $\pm$ 32.3) ppb of the CEA $\leq$ 5 ng/mL group were significantly greater than the CEA > 5 ng/mL group ($p$-value < 0.05).

**Table 2.** Student *t*-test for Age, BMI, WBC, AST, ALT, Creatinine, Air pollution in 2 different carcinoembryonic antigen (CEA) level groups.

| Factors | CEA $\leq$ 5 ng/mL Mean (SD) | CEA > 5 ng/mL Mean (SD) | *p*-Value |
|---|---|---|---|
| Age | 54.2 (5.9) | 55.8 (5.7) | <0.001 |
| BMI ($kg/m^2$) | 24.7 (3.8) | 25.1 (3.7) | 0.197 |
| WBC (/$\mu L$) | 5897.1 (1470.0) | 6437.1 (1633.2) | <0.001 |
| AST (U/L) | 28.0 (28.9) | 35.3 (33.7) | 0.001 |
| ALT (U/L) | 29.3 (20.0) | 35.8 (25.5) | <0.001 |
| Creatinine (mg/dL) | 1.2 (0.2) | 1.2 (0.2) | 0.09 |
| $SO_2$ (ppb) | 8.2 (4.8) | 7.8 (5.1) | 0.187 |
| $NO_2$ (ppb) | 25.7 (12.4) | 22.7 (11.1) | <0.001 |
| $O_3$ (ppm) | 28.2 (12.4) | 27.6 (13.3) | 0.585 |
| CO (ppm) | 0.6 (0.2) | 0.6 (0.2) | 0.095 |
| $PM_{10}$ ($\mu g/m^3$) | 68.7 (32.3) | 62.2 (28.8) | 0.001 |

Table 3 compares whether there were significant differences in CEA concentration between different groups, including gender, smoking, drinking, chewing betel nut and whether there were abnormal lung manifestations on the chest X-ray. Comparing men and women, it was found that the blood concentration of CEA in men (2.1 $\pm$ 3.3 ng/mL) was significantly higher than that in women (1.4 $\pm$ 2.6 ng/mL). The blood concentration of CEA in smokers (2.5 $\pm$ 1.7 ng/mL) was significantly higher than that of non-smokers (1.6 $\pm$ 3.2 ng/mL). The CEA blood concentration of drinkers (2.1 $\pm$ 1.6 ng/mL) was significantly higher than that of non-drinkers (1.6 $\pm$ 3.2 ng/mL). The CEA blood concentration of betel nut chewers (2.5 $\pm$ 1.8 ng/mL) was significantly higher than those that did not chew betel nuts (1.7 $\pm$ 3.0 ng/mL). In chest X-ray examinations, the CEA concentration (1.8 $\pm$ 3.6 ng/mL) for patients with abnormal lung manifestations was also significantly higher than that of those with normal or other problems (1.6 $\pm$ 2.7 ng/mL) ($p$-value < 0.05).

**Table 3.** Carcinoembryonic antigen (CEA) levels (ng/mL) in different groups: gender, tobacco, alcohol, betel nut, and chest X-ray finding.

| | Mean (SD) of CEA | *p*-Value [1] |
|---|---|---|
| Gender | | |
| Male | 2.1 (3.3) | <0.001 |
| Female | 1.4 (2.6) | |
| Tobacco | | |
| Yes | 2.5 (1.7) | <0.001 |
| No | 1.6 (3.2) | |
| Alcohol | | |
| Yes | 2.1 (1.6) | <0.001 |
| No | 1.6 (3.2) | |
| Betel nut | | |
| Yes | 2.5 (1.8) | <0.001 |
| No | 1.7 (3.0) | |
| Chest X ray | | |
| Minor abnormal lung | 1.8 (3.6) | 0.008 |
| Normal | 1.6 (2.7) | |

[1] Student *t*-test.

Table 4 used Pearson correlation to compare the correlation between CEA concentration and five air pollutants. The correlation between CEA and these five air pollutants was relatively low, with correlation coefficients of 1–3% (0.01–0.03). Among the five air pollutants, the correlation between PM10 and $NO_2$, $O_3$, CO was found to be relatively high.

**Table 4.** Pearson correlation of carcinoembryonic antigen (CEA) and air pollutants.

|  | CEA | $SO_2$ | $NO_2$ | $O_3$ | CO | $PM_{10}$ |
|---|---|---|---|---|---|---|
| CEA | 1 | −0.020 * | −0.014 | 0.028 * | 0.004 | −0.018 |
| $SO_2$ |  | 1 | 0.208 * | −0.384 ** | 0.463 ** | 0.061 * |
| $NO_2$ |  |  | 1 | 0.523 ** | 0.836 ** | 0.826 ** |
| $O_3$ |  |  |  | 1 | 0.415 ** | 0.800 ** |
| CO |  |  |  |  | 1 | 0.712 ** |
| PM10 |  |  |  |  |  | 1 |

* $p < 0.05$, ** $p < 0.001$.

Table 5 shows the results of multiple linear regressions, to understand the relationship between CEA concentrations and gender, age, BMI, white blood cell concentration, ALT, chest X-ray, smoking, drinking, betel nut, and air pollutants. Model 1 examines the relationship between four air pollutants and the concentration of CEA. When PM10 was not considered, it was found that there was a significant correlation between male, age, white blood cell, smoking, and CEA ($p < 0.05$). After controlling for other variables, CEA and the air pollutants CO and $O_3$ were significantly correlated: every increase of 1 ppm $O_3$ would significantly increase the mean differences of the CEA blood concentration by 0.005 ng/mL; every increase of 1 ppm CO would significantly reduce the mean differences of the CEA blood concentration by 0.412 ng/mL.

**Table 5.** Multiple linear regression across carcinoembryonic antigen (CEA) according to gender, age, BMI, WBC, ALT, chest X-ray, tobacco, alcohol, betel nut, air pollution.

|  | Model 1 | | | Model 2 | | |
|---|---|---|---|---|---|---|
|  | β | S.E | *p*-Value | β | S.E | *p*-Value |
| Constant | −0.319 | 0.237 | 0.179 | −0.313 | 0.237 | 0.187 |
| Gender |  |  |  |  |  |  |
| Female | Reference |  |  | Reference |  |  |
| Male | 0.314 | 0.046 | <0.001 | 0.315 | 0.046 | <0.001 |
| Age | 0.026 | 0.004 | <0.001 | 0.026 | 0.004 | <0.001 |
| BMI (kg/m$^2$) | −0.007 | 0.005 | 0.214 | −0.007 | 0.005 | 0.215 |
| WBC (/μL) | $6.39 \times 10^{-5}$ | <0.001 | <0.001 | $6.41 \times 10^{-5}$ | <0.001 | <0.001 |
| ALT (U/L) | 0.001 | 0.001 | 0.161 | 0.001 | 0.001 | 0.161 |
| Chest X ray |  |  |  |  |  |  |
| Normal | Reference |  |  | Reference |  |  |
| Minor Abnormal in lung | 0.061 | 0.047 | 0.193 | 0.060 | 0.047 | 0.196 |
| Tobacco |  |  |  |  |  |  |
| No | Reference |  |  | Reference |  |  |
| Yes | 0.637 | 0.062 | <0.001 | 0.638 | 0.062 | <0.001 |
| Alcohol |  |  |  |  |  |  |
| No | Reference |  |  | Reference |  |  |
| Yes | 0.112 | 0.059 | 0.058 | 0.112 | 0.059 | 0.059 |
| Betel nut |  |  |  |  |  |  |
| No | Reference |  |  | Reference |  |  |
| Yes | 0.219 | 0.122 | 0.075 | 0.219 | 0.123 | 0.073 |
| Air pollution |  |  |  |  |  |  |
| $SO_2$ (ppb) | 0.006 | 0.005 | 0.260 | 0.007 | 0.005 | 0.234 |
| $NO_2$ (ppb) | 0.007 | 0.004 | 0.081 | 0.006 | 0.005 | 0.190 |
| $O_3$ (ppm) | 0.005 | 0.003 | 0.040 | 0.003 | 0.003 | 0.304 |
| CO (ppm) | −0.412 | 0.180 | 0.022 | −0.455 | 0.187 | 0.015 |
| $PM_{10}$ (μg/m$^3$) |  |  |  | 0.002 | 0.002 | 0.420 |

Model 2 was to observe the relationship between five air pollutants and CEA concentration after adding PM10: men, age, white blood cells, smoking still had a significant

correlation with CEA; and for air pollutants, every 1 ppm increase in CO would significantly reduce the mean differences of the CEA blood concentration by 0.455 ng/mL.

The VIF values in model 1 were all less than 5; while in model 2, the VIF value for $PM_{10}$ was the largest, 6.14, which was considered the two models may be acceptable. Furthermore, the collinearity diagnostics among these four air pollutants in model 1, and five air pollutants in model 2 all had eigenvalues less than 9.878, which meant these two models had little trouble on collinearity of air pollutants themselves.

## 4. Discussion

The strength of our study was that it is the first empirical study of the association between serum CEA levels and air pollutants for residential locations, as far as we are aware. In the present study, we found that serum CEA levels were associated with air pollutants, which showed that every increase of 1 ppm $O_3$ would significantly increase the mean differences of the serum CEA concentration by 0.005 ng/mL. The gas $O_3$ is an agent associated with oxidant stress and low-grade inflammation [1,34]. Our results support the possibility that serum CEA levels were closely related to subclinical low-grade inflammation that would be induced by air pollution.

Some recent evidence has indicated that elevated serum CEA levels are also related to metabolic syndrome, non-alcoholic fatty liver disease, atherosclerosis, and arterial stiffness, which are conditions closely linked to chronic low-grade inflammation. Lee et al. [31] reported that serum CEA is positively associated with metabolic syndrome in female Korean non-smokers. From the same cohort of 200 healthy Korean adults, CEA was shown to be positively associated with non-alcoholic fatty liver disease [35]. Likewise, Esteghamati et al. [36] showed that metabolic syndrome and type 2 diabetes are more prevalent in subjects with high CEA levels. In 4181 Japanese men aged 21–89 years, serum CEA was independently associated with carotid atherosclerosis [30]. In addition, Bae et al. [37] reported that CEA levels are related to arterial stiffness as measured by brachial-ankle pulse wave velocity in healthy Korean adults. The pathophysiologic and clinical significance of these findings, however, requires further study. This association suggests that other mechanisms, at least in part, independent of inflammation may be at play. Some previous studies have shown that serum CEA was associated with increased insulin resistance [38] and metabolic syndrome [39], which are closely associated with an increased risk of CVD. The pathophysiologic and clinical significance of these findings, however, requires further study.

On the other hand, some studies have documented the relationship between CEA and metabolic syndrome, diabetes, and insulin resistance [40,41]. A systematic review and meta-analysis showed a higher level of $PM_{2.5}$ exposure was associated with higher type 2 diabetes incidence (per 10 μg/m$^3$) increase in the concentration of $PM_{2.5}$; meanwhile, 12 studies reported the effects of $NO_2$ exposure on type 2 diabetes prevalence and the pooled estimates suggested a positive relationship (standardized OR = 1.05, 95%CI: 1.03, 1.08) via meta-analysis [40]. The serum CEA was positively associated with metabolic syndrome and pre-diabetes in the previous studies [31,38,39]. Thus, the serum CEA could be associated with air pollution. In addition, a study used the Taiwan National Health Insurance Research Database (NHIRD) and concluded that long-term exposure to ambient $O_3$ and $SO_2$ was associated with a higher risk of developing type 2 diabetes for the Taiwanese population [41]. The NHIRD study used the same resource of air pollution as ours. Our study found $O_3$ was positively associated with the serum CEA. Thus, a rational association could be concluded.

In addition, with adjustment of potential confounders, we found that every increase of 1 ppm CO would significantly reduce the mean differences of the blood concentration of CEA by 0.455 ng/mL. Carbon monoxide (CO) is a common environmental pollutant. Studies have shown that under controlled conditions to simulate daily urban environmental pollution, four weeks of CO exposure would accelerate the deterioration of heart failure caused by myocardial infarction [6,34,42]. However, some studies have suggested that

exogenous CO at lower concentrations may have beneficial anti-inflammatory effects under certain circumstances [9–11]. A 0.5 ppm increase of CO was associated with a 3.5% and 2.4% decrease in cardiovascular and total mortality, respectively [14–16]. A protective effect of CO on daily mortality was also indicated in a study of 10 U.S. cities: 1 ppm increase of CO was associated with a 0.7% decrease in daily mortality [43]. It may be a reason why increased CO related to decreasing serum CEA levels. However, further research should be needed to evaluate the overall health effects of exposure to low environmental CO levels.

It may also be modestly elevated in several non-neoplastic conditions [27–29]. Of these, aging and cigarette smoking appear to be major determinants of elevated CEA levels. Touitou et al. [23] reported that elderly subjects in apparent well health had higher CEA levels; while Fukuda et al. [24] showed that serum CEA levels are higher in male smokers (3.11 ng/mL) than in male non-smokers (2.14 ng/mL). This research found similar results. In addition, we used regression, a multiple covariate control, to adjust the confounders, such as smoking, alcohol drinking, in the regression models; thus, the air pollutants affected CEA levels could be identified.

Particulate matter (PM) is a factor that causes diseases and is widely studied. However, the main gas pollutants (CO, $NO_2$, CO and $SO_2$) also play an important role, reminding us that air pollutants are a key consideration. Regarding sulphur dioxide ($SO_2$), it is one of the most common causes of air pollution; especially for children and older elders, $SO_2$ is easily harmful to the respiratory tract, and may also aggravate heart disease and lung disease, and is more likely to increase the severity of asthma. At high exposure concentrations (above 50 ppm), it is easier to damage the larynx, trachea, and bronchi [44]. The concentration of sulphur dioxide in this study averaged 8.2 ($\pm$4.8) ppb. It may be exposed to a lower concentration and less chance of acute damage to the respiratory tract. Therefore, it is observed that there is no significant relationship with CEA. As for the part containing nitrogen oxides, the most toxic ones are nitric oxide and nitrogen dioxide. Nitrogen oxides in low concentrations (15 to 25 ppm) can irritate the lower respiratory tract and may cause mild shortness of breath and coughing [44,45]. The concentration of $NO_2$ in this study was 25.6 ($\pm$12.4) ppb, which is lower than the range that may affect the organs, so the respiratory tract is less likely to be exposed to it, and it may be less significantly related to CEA.

Our study has some limitations. First, the sample came from the public health examination data of volunteers in southern Taiwan, although we randomly selected the four metropolitan districts and used the stratified proportion of the population of randomly selected districts. The volunteer effect, a possible selection bias could affect the result. However, with the high population mobility in Taiwan, we think data from volunteers in southern Taiwan could at least partially reflect the overall impact of the air pollutants.

Second, the monitoring concentration of air pollutants was the average of the two days, before and the same day, to participate in the health examination. It was still a cross-sectional study. The causal relationship was weaker, compared to long-term follow-up studies. In addition, as the data source was a one-time test, it cannot be obtained to know the long-term effects of exposure to air pollutants on CEA concentrations.

Third, to evaluate inflammatory conditions, only one measurement of leukocyte levels was performed, making it difficult to determine whether the increased leukocyte count was reflective of acute inflammation or chronic inflammation. To reduce these errors, we excluded subjects with the criteria leukocyte counts <3000 cells/μL or >10,000 cells/μL.

Fourth, we did not use some computing methods, such as geographic information systems (GIS), to estimate the exposure. The air pollutants concentrations used the measurements from the monitoring data of the nearest air monitoring station at the subject's address. Then, there may be deviations from the actual concentrations. However, the deviations would be small, because there was at least one air monitoring station in each district in the metropolitan area of Kaohsiung City. Thus, our finding was still useful to future researchers using GIS with air pollution models.

Finally, because the study subjects were volunteers who received their health examinations in a single hospital and appeared to be slightly healthier than most hospital-based studies, However the study population may really represent the general population of southern Taiwan.

Despite these limitations, this is the first study to identify a positive association between CEA and air pollutants in Taiwan. We consider the non-malignant conditions related to chronic inflammation as due to the modestly increased serum CEA levels caused by air pollutants.

## 5. Conclusions

We identified a significant association between serum CEA and the levels of the air pollutant $O_3$ and CO in the community population. Our findings suggest that serum CEA may be a meaningful marker of chronic low-grade inflammation in clinical practice. In conclusion, the changes in CEA concentration and short-term exposure to air pollutants $O_3$ and CO were found to have a significant relationship. The results showed that every increase of 1 ppm $O_3$ significantly increased the mean differences of the CEA blood concentration by 0.005 ng/mL. Each increase of 1 ppm CO significantly reduced the mean differences of the CEA in blood concentration by 0.455 ng/mL, but its mechanism is still unclear. Long-term air pollution exposure and changes in CEA concentration still need to be further evaluated.

**Author Contributions:** Conceptualization, Y.-J.F. and H.-Y.C.; methodology, H.-Y.C.; validation, Y.-J.F., L.J.-H.L. and H.-Y.C.; formal analysis, K.-H.L.; investigation, Y.-J.F., L.J.-H.L., P.-S.F. and C.-C.Y.; resources, H.-Y.C.; data curation, Y.-J.F., P.-S.F. and C.-C.Y.; writing—original draft preparation, Y.-J.F. and L.J.-H.L.; writing—review and editing, H.-Y.C.; supervision, H.-Y.C.; project administration, K.-H.L. and H.-Y.C.; funding acquisition, H.-Y.C. All authors have read and agreed to the published version of the manuscript.

**Funding:** This research was funded by Ministry of Science and Technology, grant number NSC99-2632-B-037-001-MY3, and Kaohsiung Medical University Hospital (KMUH110-0T01), and Kaohsiung Medical University (110KK037).

**Institutional Review Board Statement:** The study, with protocol number of KMUH-IRB-990206, was conducted according to the guidelines of the Declaration of Helsinki and approved by the Institutional Review Board of Kaohsiung Medical University Chung-Ho Memorial Hospital.

**Informed Consent Statement:** Informed consent was obtained from all subjects involved in the study.

**Acknowledgments:** We thank the participants for their cooperation, and the help from the Health Bureau of Kaohsiung Municipality. This work was supported partially by the Research Center for Environmental Medicine, Kaohsiung Medical University, Kaohsiung City, Taiwan from The Featured Areas Research Center Program within the framework of the Higher Education Sprout Project by the Ministry of Education (MOE) in Taiwan and by Kaohsiung Medical University Research Center Grant (KMU-TC111A01 and KMUTC111IFSP01).

**Conflicts of Interest:** The authors declare no conflict of interest.

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
