# Peer review of "The Association of Carcinoembryonic Antigen (CEA) and Air Pollutants—A Population-Based Study"

_atmosphere, doi:10.3390/atmos13030466_

Round 1

Reviewer 1 Report

  1. “health examiners” (line 160-163) should be the doctors to examine the patients . “health examiners” are not the people who are examined. The subject to be examined is “examinee”.
  2. “The air pollution concentrations were detected by the air monitoring station nearest to participants’ address” (line 137). How to explain “nearest”? How far for every participant? I believe it is not equidistant. The participant’s address may be surrounded with several air monitoring stations unequally or equally, so the air pollutants concentrations’ interpolations at the participant’s address should be computed with  GIS (geographic information system).

Reviewer 2 Report

Fang and cols. evaluated the hypothesis of association (short-term) between ambient air pollution (CO, O3, PM10, NO2, and SO2) and CEA concentrations in a sample of subjects from the catchment area of the Kaohsiung Municipal Siaogang Hospital (without history of cancer, respiratory, renal, hepatobiliary, thyroid, or rheumatologic diseases), in the context of a cross-sectional design.

Comments

Methods.

  • It is not clear what is the sample frame of the study. The description of the sampling process suggests that all residents of Kaohsiung were part of the frame (lines 119-121), however, only in the discussion section (lines 284-285) it becomes clear that participants were drawn from the catchment area of one hospital en the municipality.
  • Please clarify whether anthropometric indexes (weight and height) were self-reported (as suggested in line 124) or measured by the investigators.
  • I suggest providing a more detailed description of the exposure, specifically, regarding the local air quality network (for example, number of monitors, density in the study area, etc.) so the readers will have a better idea of concentrations' spatial and temporal variability.
  • Considering that air pollutants' concentrations are highly correlated (the results support this), did the authors estimate the variance inflation factor (VIF) as part of model building? I suggest describing and reporting estimates for the models' goodness of fit and declaring whether a statistical criterion was used to define and include confounders in the models.
  • The correct interpretation of Pearson's correlation coefficients supposes that variables are normally distributed and have a linear (rectilinear) relationship. Where these premises tested? If so, did any variable need to be transformed (for instance, log-transformed)?

Results.

  • This section is well presented; however, the authors should check for consistency of concentration estimates for CO and SO2 between tables 1 and 2 (also reflected in lines 175-176 and 182-184 of this section and line 275 of the discussion).
  • Please check on the p-value for the correlation coefficient between CO and SO2 in table 4 (the magnitude and statistical significance of correlations between CO-NO2 [0.836**] and NO2-SO2 [0.208**] suggest there is a typo).
  • I also suggest to interpreting regression coefficients in terms of mean differences instead of changes of Y by changes in X (for example, lines 214-216) which might be misleading.

Discussion/conclusion.

  • In this section the authors orderly approach and contextualize each of the main findings of the study.
  • Although the relations between CEA and metabolic syndrome, diabetes, and insulin resistance are well documented the authors do not discuss the potential implication(s) of not considering any of those health conditions in the analysis, specifically, residual confounding given the known relationship between exposure to air pollution and them (for example: Liu et al. 2019 [doi: 10.1016/j.envpol.2019.06.033] and Li et al. 2021 [doi: 10.1016/j.envres.2020.110624]).
  • No strengths are cited: This study surely have some that I prompt the authors to declare. In terms of limitations: 1) Selection bias is discussed in terms of external but not internal validity (i.e., whether there was a differential probability of inclusion of participants based on their exposure and outcome); however, considering the sampling strategy and that the CEA concentration was not an eligibility criterion, the risk of this bias could be considered low. 2) Information bias is not discussed in this section although considering the ascertainment of the exposure, at least there exists the possibility of non-differential classification. 3) Confounding, specifically, residual confounding and its implication to the interpretation of results is missing (for example, in the case of diabetes).
  • In the conclusion I suggest rephrasing the interpretation of the main findings in terms of mean differences instead of changes of Y by changes in X at the light of the study design. 

Reviewer 3 Report

This is a cross-sectional study to evaluate the association of serum CEA levels and the air pollutants. The authors found that the serum CEA concentrations and short-term air pollutant O3, CO exposure had a significant relationship. Although the study has novelty and the manuscript was well presented, the reviewer was still concerned about the following issues.

Major comments:

  1. As the authors mentioned in the Introduction, the serum CEA levels may be influenced by renal function, liver function, type 2 DM and various adenocarcinomas. Please rule out or adjust all these confounding factors.
  2. It was stated in the Lines 203-204, Page 6: “The correlation between CEA and these five air pollutants is relatively low, with correlation coefficients of 1-3.” Usually, the correlation coefficient is between -1 and 1, please clarify it.
  3. Please use past tense verbs for the findings of this study in the Abstract, Results and Discussion.

Minor comments:

  1. “Pearson correlation analysis is used to evaluate the correlation between CEA air pollution (Lines 150-151, Page 3).” is better revised as “Pearson correlation analysis was used to evaluate the correlation between CEA and air pollution.”
  2. “Use Multiple liner regression to analyze the variables, and establish two sets of models for discussion (Lines 152-153, Page3).” Is better revised as “We used multiple liner regression to analyze the variables, and established two sets of models for comparison.”
  3. “In routine blood tests, the average value of white blood cells was 911.1 (±1477.8)/μL (Lines 165-166, Page 4).” should be corrected as “In routine blood tests, the average value of white blood cells was 1 (±1477.8)/μL;”.
  4. Double statements were noted in the Lines 160-164, Page 4 as “Table 1 shows the basic characters of health examiners. The number of health examiners was 4188 (43.1%) males and 5540 (56.9%) females. The average (± standard deviation) age is 54.3 (±5.9) years old. Table 1 shows the basic characters of health examiners. The number of health examiners was 4188 (43.1%) males and 5540 (56.9%) females. The aver-163 age (± standard deviation) age is 54.3 (±5.9) years old.
  5. The “m2” in Line 165, Page 4 should be revised as “m2”.
  6. “The results showed that every increase of 1 ppm O3 will significantly increase the CEA blood con-centration by 0.005 ng/mL. Each increase of 1 ppm CO will significantly reduce the CEA blood concentration by 0.455 ng/mL (Lines 313-315, Page 8)” is better revised as “The results showed that every increase of 1 ppm 313 O3 may significantly increase the CEA blood con-centration by 0.005 ng/mL. Each increase 314 of 1 ppm CO may significantly reduce the CEA blood concentration by 0.455 ng/mL.” It is very bizarre to use future tense verbs here.

Round 2

Reviewer 1 Report

It is suggested to explain in the limitation column, that  the air pollutants concentrations interpolations at the subject’s address had not been computed with GIS (geographic information system). The air pollutants concentrations used for every subject is the monitoring data of the nearest air monitoring station, and there may be deviations from the actual concentrations.
